# Neural Networks with Cheap Differential Operators

**Ricky T. Q. Chen,  David Duvenaud**
University of Toronto, Vector Institute
{rtqichen,duvenaud}@cs.toronto.edu

## Abstract

Gradients of neural networks can be computed efficiently for any architecture, but some applications require differential operators with higher time complexity. We describe a family of restricted neural network architectures that allow efficient computation of a family of differential operators involving dimension-wise derivatives, used in cases such as computing the divergence. Our proposed architecture has a Jacobian matrix composed of diagonal and hollow (non-diagonal) components. We can then modify the backward computation graph to extract dimension-wise derivatives efficiently with automatic differentiation. We demonstrate these cheap differential operators for solving root-finding subproblems in implicit ODE solvers, exact density evaluation for continuous normalizing flows, and evaluating the Fokker–Planck equation for training stochastic differential equation models.

## 1   Introduction

Artificial neural networks are useful as arbitrarily-flexible function approximators (Cybenko, 1989; Hornik, 1991) in a number of fields. However, their use in applications involving differential equations is still in its infancy. While many focus on the training of black-box neural nets to approximately represent solutions of differential equations (e.g., Lagaris et al. (1998); Tompson et al. (2017)), few have focused on designing neural networks such that differential operators can be efficiently applied.

In modeling differential equations, it is common to see differential operators that require only dimension-wise or element-wise derivatives, such as the Jacobian diagonal, the divergence (ie. the Jacobian trace), or generalizations involving higher-order derivatives. Often we want to compute these operators when evaluating a differential equation or as a downstream task. For instance, once we have fit a stochastic differential equation, we may want to apply the Fokker–Planck equation (Risken, 1996) to compute the probability density, but this requires computing the divergence and other differential operators. The Jacobian diagonal can also be used in numerical optimization schemes such as the accelerating fixed-point iterations, where it can be used to approximate the full Jacobian while maintaining the same fixed-point solution.

In general, neural networks do not admit cheap evaluation of arbitrary differential operators. If we view the evaluation of a neural network as traversing a computation graph, then reverse-mode automatic differentiation–a.k.a backpropagation–traverses the exact same set of nodes in the reverse direction (Griewank and Walther, 2008; Schulman et al., 2015). This allows us to compute what is mathematically equivalent to vector-Jacobian products with asymptotic time cost equal to that of the forward evaluation. However, in general, the number of backward passes—ie. vector-Jacobian products—required to construct the full Jacobian for unrestricted architectures grows linearly with the dimensionality of the input and output. Unfortunately, this is also true for extracting the diagonal elements of the Jacobian needed for differential operators such as the divergence.

In this work, we construct a neural network in a manner that allows a family of differential operators involving dimension-wise derivatives to be cheaply accessible. We then modify the backward computation graph to efficiently compute these derivatives with a single backward pass.

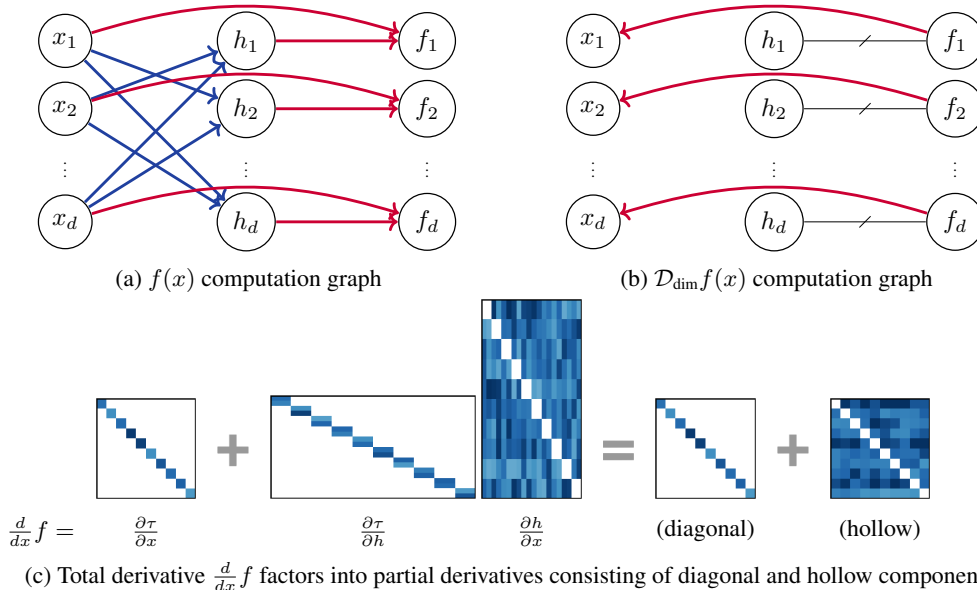

(a) $f(x)$ computation graph

(b) $\mathcal{D}_{\text{dim}}f(x)$ computation graph

$$\frac{d}{dx}f = \qquad \frac{\partial \tau}{\partial x} \qquad\qquad \frac{\partial \tau}{\partial h} \qquad \frac{\partial h}{\partial x} \qquad = \qquad \text{(diagonal)} \qquad \text{(hollow)}$$

(c) Total derivative $\frac{d}{dx}f$ factors into partial derivatives consisting of diagonal and hollow components.

Figure 1: (a) Visualization of HollowNet's computation graph, which is composed of a conditioner network (blue) and a transformer network (red). Each line represents a non-linear dependency. (b) We can modify the backward pass to retain only dimension-wise dependencies, which exist only through the transformer network. (c) The connections are designed so we can easily factor the full Jacobian matrix $\frac{d}{dx}f$ into diagonal and hollow components (visualized for $d_h$=2).

## 2 HollowNet: Segregating Dimension-wise Derivatives

Given a function $f : \mathbb{R}^d \to \mathbb{R}^d$, we seek to obtain a vector containing its *dimension-wise $k$-th order derivatives*,

$$\mathcal{D}_{\text{dim}}^k f := \begin{bmatrix} \frac{\partial^k f_1(x)}{\partial x_1^k} & \frac{\partial^k f_2(x)}{\partial x_2^k} & \cdots & \frac{\partial^k f_d(x)}{\partial x_d^k} \end{bmatrix}^T \in \mathbb{R}^d \qquad (1)$$

using only $k$ evaluations of automatic differentiation regardless of the dimension $d$. For notational simplicity, we denote $\mathcal{D}_{\text{dim}} := \mathcal{D}_{\text{dim}}^1$.

We first build the forward computation graph of a neural network such that the the Jacobian matrix is composed of a diagonal and a hollow matrix, corresponding to dimension-wise partial derivatives and interactions between dimensions respectively. We can efficiently apply the $\mathcal{D}_{\text{dim}}^k$ operator by disconnecting the connections that represent the hollow elements of the Jacobian in the backward computation graph. Due to the structure of the Jacobian matrix, we refer to this type of architecture as a *HollowNet*.

### 2.1 Building the Computation Graph

We build the HollowNet architecture for $f(x)$ by first constructing hidden vectors for each dimension $i$, $h_i \in \mathbb{R}^{d_h}$, that don't depend on $x_i$, and then concatenating them with $x_i$ to be fed into an arbitrary neural network. The combined architecture sets us up for modifying the backward computation graph for cheap access to dimension-wise operators. We can describe this approach as two main steps, borrowing terminology from Huang et al. (2018):

1. **Conditioner.** $h_i = c_i(x_{-i})$ where $c_i : \mathbb{R}^{d-1} \to \mathbb{R}^{d_h}$ and $x_{-i}$ denotes the vector of length $d-1$ where the $i$-th element is removed. All states $\{h_i\}_{i=1}^d$ can be computed in parallel by using networks with masked weights, which exist for both fully connected (Germain et al., 2015) and convolutional architectures (Oord et al., 2016).

2. **Transformer.** $f_i(x) = \tau_i(x_i, h_i)$ where $\tau_i : \mathbb{R}^{d_h+1} \to \mathbb{R}$ is a neural network that takes as input the concatenated vector $[x_i, h_i]$. All dimensions of $f_i(x)$ can be computed in parallel if the $\tau_i$'s are composed of matrix-vector and element-wise operations, as is standard in deep learning.

**Relaxation of existing work.** This family of architectures contains existing special cases, such as Inverse (Kingma et al., 2016), Masked (Papamakarios et al., 2017) and Neural (Huang et al., 2018) Autoregressive Flows, as well as NICE and Real NVP (Dinh et al., 2014, 2016). Notably, existing works focus on constraining $f(x)$ to have a triangular Jacobian by using a conditioner network with a specified ordering. They also choose $\tau_i$ to be invertible. In contrast, we relax both constraints as they are not required for our application. We compute $h = c(x)$ in parallel by using two masked autoregressive networks (Germain et al., 2015).

**Expressiveness.** This network introduces a bottleneck in terms of expressiveness. If $d_h \geq d - 1$, then it is at least as expressive as a standard neural network, since we can simply set $h_i = x_{-i}$ to recover the same expressiveness as a standard neural net. However, this would require $\mathcal{O}(d^2)$ total number of hidden units for each evaluation, resulting in having similar cost—though with better parallelization still—to naïvely computing the full Jacobian of general neural networks with $d$ AD calls. For this reason, we would like to have $d_h \ll d$ to reduce the amount of compute in evaluating our network. It is worth noting that existing works that make use of masking to parallelize computation typically use $d_h = 2$ which correspond to the scale and shift parameters of an affine transformation $\tau$ (Kingma et al., 2016; Papamakarios et al., 2017; Dinh et al., 2014, 2016).

## 2.2 Splicing the Computation Graph

Here, we discuss how to compute dimension-wise derivatives for the HollowNet architecture. This procedure allows us to obtain the exact Jacobian diagonal at a cost of only one backward pass whereas the naïve approach would require $d$.

A single call to reverse-mode automatic differentiation (AD)—ie. a single backward pass—can compute vector-Jacobian products.

$$v^T \frac{df(x)}{dx} = \sum_i v_i \frac{df_i(x)}{dx} \tag{2}$$

By constructing $v$ to be a one-hot vector—ie. $v_i = 1$ and $v_j = 0 \; \forall j \neq i$—then we obtain a single row of the Jacobian $df_i(x)/dx$ which contains the dimension-wise derivative of the $i$-th dimension. Unfortunately, to obtain the full Jacobian or even $\mathcal{D}_{\mathrm{dim}}f$ would require $d$ AD calls.

Now suppose the computation graph of $f$ is constructed in the manner described in 2.1. Let $\widehat{h}$ denote $h$ but with the backward connection removed, so that AD would return $\partial \widehat{h}/\partial x_j = 0$ for any index $j$. This kind of computation graph modification can be performed with the use of `stop_gradient` in Tensorflow (Abadi et al., 2016) or `detach` in PyTorch (Paszke et al., 2017). Let $\widehat{f}(x) = \tau(x, \widehat{h})$, then the Jacobian of $\widehat{f}(x)$ contains only zeros on the off-diagonal elements.

$$\frac{\partial \widehat{f}_i(x)}{\partial x_j} = \frac{\partial \tau_i(x_i, \widehat{h}_i)}{\partial x_j} = \begin{cases} \frac{\partial f_i(x)}{\partial x_i} & \text{if } i = j \\ 0 & \text{if } i \neq j \end{cases} \tag{3}$$

As the Jacobian of $\widehat{f}(x)$ is a diagonal matrix, we can recover the diagonals by computing a vector-Jacobian product with a vector with all elements equal to one, denoted as $\mathbb{1}$.

$$\mathbb{1}^T \frac{\partial \widehat{f}(x)}{\partial x} = \mathcal{D}_{\mathrm{dim}}\widehat{f} = \begin{bmatrix} \frac{\partial f_1(x)}{\partial x_1} & \cdots & \frac{\partial f_d(x)}{\partial x_d} \end{bmatrix}^T = \mathcal{D}_{\mathrm{dim}}f \tag{4}$$

The higher orders $\mathcal{D}_{\mathrm{dim}}^k$ can be obtained by $k$ AD calls, as $\widehat{f}_i(x)$ is only connected to the $i$-th dimension of $x$ in the computation graph, so any differential operator on $\widehat{f}_i$ only contains the dimension-wise connections. This can be written as the following recursion:

$$\mathbb{1}^T \frac{\partial \mathcal{D}_{\mathrm{dim}}^{k-1}\widehat{f}(x)}{\partial x} = \mathcal{D}_{\mathrm{dim}}^k\widehat{f}(x) = \mathcal{D}_{\mathrm{dim}}^k f(x) \tag{5}$$

As connections have been removed from the computation graph, backpropagating through $\mathcal{D}_{\mathrm{dim}}^k\widehat{f}$ would give erroneous gradients as the connections between $f_i(x)$ and $x_j$ for $j \neq i$ were severed. To ensure correct gradients, we must reconnect $\widehat{h}$ and $h$ in the backward pass,

$$\frac{\partial \mathcal{D}_{\mathrm{dim}}^k\widehat{f}}{\partial w} + \frac{\partial \mathcal{D}_{\mathrm{dim}}^k\widehat{f}}{\partial \widehat{h}} \frac{\partial h}{\partial w} = \frac{\partial \mathcal{D}_{\mathrm{dim}}^k f}{\partial w} \tag{6}$$

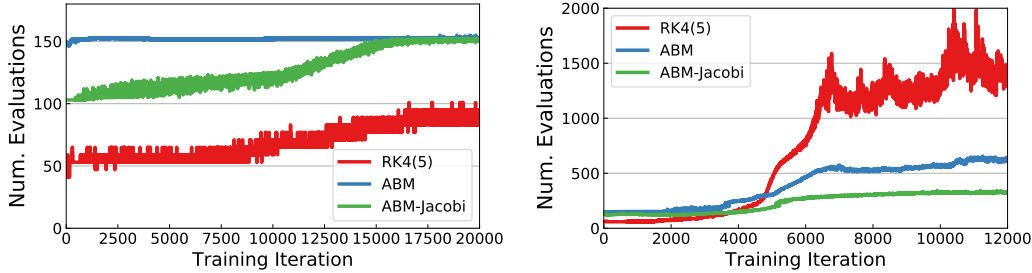

(a) Explicit solver is sufficient for nonstiff dynamics.

(b) Training may result in stiff dynamics.

Figure 2: Comparison of ODE solvers as differential equation models are being trained. (a) Explicit methods such as RK4(5) are generally more efficient when the system isn't too stiff. (b) However, when a trained dynamics model becomes stiff, predictor-corrector methods (ABM & ABM-Jacobi) are much more efficient. In difficult cases, the Jacobi-Newton iteration (ABM-Jacobi) uses significantly less evaluations than functional iteration (ABM). A median filter with a kernel size of 5 iterations was applied prior to visualization.

where $w$ is any node in the computation graph. This gradient computation can be implemented as a custom backward procedure, which is available in most modern deep learning frameworks.

Equations (4), (5), and (6) perform computations on only $\widehat{f}$—shown on the left-hand-sides of each equation—to compute dimension-wise derivatives of $f$—the right-hand-sides of each equation. The number of AD calls is $k$ whereas naïve backpropagation would require $k \cdot d$ calls. We note that this process can only be applied to a single HollowNet, since for a composition of two functions $f$ and $g$, $\mathcal{D}_{\mathrm{dim}}(f \circ g)$ cannot be written solely in terms of $\mathcal{D}_{\mathrm{dim}}f$ and $\mathcal{D}_{\mathrm{dim}}g$.

In the following sections, we show how efficient access to the $\mathcal{D}_{\mathrm{dim}}$ operator provides improvements in a number of settings including (i) more efficient ODE solvers for stiff dynamics, (ii) solving for the density of continuous normalizing flows, and (iii) learning stochastic differential equation models by Fokker–Planck matching. *Each of the following sections are stand-alone and can be read individually.*

## 3 Efficient Jacobi-Newton Iterations for Implicit Linear Multistep Methods

Ordinary differential equations (ODEs) parameterized by neural networks are typically solved using explicit methods such as Runge-Kutta 4(5) (Hairer and Peters, 1987). However, the learned ODE can often become stiff, requiring a large number of evaluations to accurately solve with explicit methods. Instead, implicit methods can achieve better accuracy at the cost of solving an inner-loop fixed-point iteration subproblem at every step. When the initial guess is given by an explicit method of the same order, this is referred to as a predictor-corrector method (Moulton, 1926; Radhakrishnan and Hindmarsh, 1993). Implicit formulations also show up as inverse problems of explicit methods. For instance, an Invertible Residual Network (Behrmann et al., 2018) is an invertible model that computes forward Euler steps in the forward pass, but requires solving (implicit) backward Euler for the inverse computation. Though the following describes and applies HollowNet to implicit ODE solvers, our approach is generally applicable to solving root-finding problems.

The class of linear multistep methods includes forward and backward Euler, explicit and implicit Adams, and backward differentiation formulas:

$$
\begin{aligned}
y_{n+s} + a_{s-1}y_{n+s-1} + a_{s-2}y_{n+s-2} + \cdots + a_0 y_n \\
= h(b_s f(t_{n+s}, y_{n+s}) + b_{s-1}f(t_{n+s-1}, y_{n+s-1}) + \cdots + b_0 f(t_n, y_n))
\end{aligned}
\tag{7}
$$

where the values of the state $y_i$ and derivatives $f(t_i, y_i)$ from the previous $s$ steps are used to solve for $y_{n+s}$. When $b_s \neq 0$, this requires solving a non-linear optimization problem as both $y_{n+s}$ and $f(t_{n+s}, y_{n+s})$ appears in the equation, resulting in what is known as an implicit method.

Simplifying notation with $y = y_{n+s}$, we can write (7) as a root finding problem:

$$
F(y) := y - hb_s f(y) - \delta = 0
\tag{8}
$$

where $\delta$ is a constant representing the rest of the terms in (7) from previous steps. Newton-Raphson can be used to solve this problem, resulting in an iterative algorithm

$$y^{(k+1)} = y^{(k)} - \left[ \frac{\partial F(y^{(k)})}{\partial y^{(k)}} \right]^{-1} F(y^{(k)}) \tag{9}$$

When the full Jacobian is expensive to compute, one can approximate using the diagonal elements. This approximation results in the Jacobi-Newton iteration (Radhakrishnan and Hindmarsh, 1993).

$$\begin{aligned} y^{(k+1)} &= y^{(k)} - [\mathcal{D}_{\dim}F(y)]^{-1} \odot F(y^{(k)}) \\ &= y^{(k)} - [\mathbb{1} - hb_s\mathcal{D}_{\dim}f(y)]^{-1} \odot (y - hb_s f(y) - \delta) \end{aligned} \tag{10}$$

where $\odot$ denotes the Hadamard product, $\mathbb{1}$ is a vector with all elements equal to one, and the inverse is taken element-wise. Each iteration requires evaluating $f$ once. In our implementation, the fixed point iteration is repeated until $||y^{(k-1)} - y^{(k)}|| / \sqrt{d} \leq \tau_a + \tau_r ||y^{(0)}||_\infty$ for some user-provided tolerance parameters $\tau_a, \tau_r$.

Alternatively, when the Jacobian in (9) is approximated by the identity matrix, the resulting algorithm is referred to as functional iteration (Radhakrishnan and Hindmarsh, 1993). Using our efficient computation of the $\mathcal{D}_{\dim}$ operator, we can apply Jacobi-Newton and obtain faster convergence than functional iteration while maintaining the same asymptotic computation cost per step.

## 3.1 Empirical Comparisons

We compare a standard Runge-Kutta (RK) solver with adaptive stepping (Shampine, 1986) and a predictor-corrector Adams-Bashforth-Moulton (ABM) method in Figure 2. A learned ordinary differential equation is used as part of a continuous normalizing flow (discussed in Section 4), and training requires solving this ordinary differential equation at every iteration. We initialized the weights to be the same for fair comparison, but the models may have slight numerical differences during training due to the amounts of numerical error introduced by the different solvers. The number of function evaluations includes both evaluations made in the forward pass and for solving the adjoint state in the backward pass for parameter updates as in Chen et al. (2018). We applied Jacobi-Newton iterations for ABM-Jacobi using the efficient $\mathcal{D}_{\dim}$ operator in both the forward and backward passes.

As expected, if the learned dynamics model becomes too stiff, RK results in using very small step sizes and uses almost 10 times the number of evaluations as ABM with Jacobi-Newton iterations. When implicit methods are used with HollowNet, Jacobi-Newton can help reduce the number of evaluations at the cost of just one extra backward pass.

## 4 Continuous Normalizing Flows with Exact Trace Computation

Continuous normalizing flows (CNF) (Chen et al., 2018) transform particles from a base distribution $p(x_0)$ at time $t_0$ to another time $t_1$ according to an ordinary differential equation $\frac{dh}{dt} = f(t, h(t))$.

$$x := x(t_1) = x_0 + \int_{t_0}^{t_1} f(t, h(t)) dt \tag{11}$$

The change in distribution as a result of this transformation is described by an instantaneous change of variables equation (Chen et al., 2018),

$$\frac{\partial \log p(t, h(t))}{\partial t} = -\text{Tr}\left( \frac{\partial f(t, h(t))}{\partial h(t)} \right) = -\sum_{i=1}^{d} [\mathcal{D}_{\dim}f]_i \tag{12}$$

If (12) is solved along with (11) as a combined ODE system, we can obtain the density of transformed particles at any desired time $t_1$.

Due to requiring $d$ AD calls to compute $\mathcal{D}_{\dim}f$ for a black-box neural network $f$, Grathwohl et al. (2019) adopted a stochastic trace estimator (Skilling, 1989; Hutchinson, 1990) to provide unbiased estimates for $\log p(t, h(t))$. Behrmann et al. (2018) used the same estimator and showed that the standard deviation can be quite high for single examples. Furthermore, an unbiased estimator of the log-density has limited uses. For instance, the IWAE objective (Burda et al., 2015) for estimating a

Table 1: Evidence lower bound (ELBO) and negative log-likelihood (NLL) for static MNIST and Omniglot in nats. We outperform CNFs with stochastic trace estimates (FFJORD), but surprisingly, our improved approximate posteriors did not result in better generative models than Sylvester Flows (as indicated by NLL). Bolded estimates are *not* statistically significant by a two-tailed $t$-test with significance level 0.05.

| Model | MNIST | | Omniglot | |
|---|---|---|---|---|
| | -ELBO $\downarrow$ | NLL $\downarrow$ | -ELBO $\downarrow$ | NLL $\downarrow$ |
| VAE (Kingma and Welling, 2013) | $86.55 \pm 0.06$ | $82.14 \pm 0.07$ | $104.28 \pm 0.39$ | $97.25 \pm 0.23$ |
| Planar (Rezende and Mohamed, 2015) | $86.06 \pm 0.31$ | $81.91 \pm 0.22$ | $102.65 \pm 0.42$ | $96.04 \pm 0.28$ |
| IAF (Kingma et al., 2016) | $84.20 \pm 0.17$ | $80.79 \pm 0.12$ | $102.41 \pm 0.04$ | $96.08 \pm 0.16$ |
| Sylvester (van den Berg et al., 2018) | $83.32 \pm 0.06$ | $\mathbf{80.22 \pm 0.03}$ | $99.00 \pm 0.04$ | $\mathbf{93.77 \pm 0.03}$ |
| FFJORD (Grathwohl et al., 2019) | $82.82 \pm 0.01$ | — | $98.33 \pm 0.09$ | — |
| Hollow-CNF | $\mathbf{82.37 \pm 0.04}$ | $\mathbf{80.22 \pm 0.08}$ | $\mathbf{97.42 \pm 0.05}$ | $\mathbf{93.90 \pm 0.14}$ |

lower bound of the log-likelihood $\log p(x) = \log \mathbb{E}_{z \sim p(z)}[p(x|z)]$ of latent variable models has the following form:

$$\mathcal{L}_{\text{IWAE-}k} = \mathbb{E}_{z_1,\dots,z_k \sim q(z|x)} \left[ \log \frac{1}{k} \sum_{i=1}^{k} \frac{p(x, z_i)}{q(z_i|x)} \right] \quad (13)$$

Flow-based models have been used as the distribution $q(z|x)$ (Rezende and Mohamed, 2015), but an unbiased estimator of $\log q$ would not translate into an unbiased estimate of this importance weighted objective, resulting in biased evaluations and biased gradients if used for training. For this reason, FFJORD (Grathwohl et al., 2019) was unable to report approximate log-likelihood values for evaluation which are standardly estimated using (13) with $k = 5000$ (Burda et al., 2015).

### 4.1 Exact Trace Computation

By constructing $f$ in the manner described in Section 2.1, we can efficiently compute $\mathcal{D}_{\text{dim}} f$ and the trace. This allows us to exactly compute (12) using a single AD call, which is the same cost as the stochastic trace estimator. We believe that using exact trace should reduce gradient variance during training, allowing models to converge to better local optima. Furthermore, it should help reduce the complexity of solving (12) as stochastic estimates can lead to more difficult dynamics.

### 4.2 Latent Variable Model Experiments

We trained variational autoencoders (Kingma and Welling, 2013) using the same setup as van den Berg et al. (2018). This corresponds to training using (13) with $k = 1$, also known as the evidence lower bound (ELBO). We searched for $d_h \in \{32, 64, 100\}$ and used $d_h = 100$ as the computational cost was not significantly impacted. We used 2-3 hidden layers for the conditioner and transformer networks, with the ELU activation function. Table 1 shows that training CNFs with exact trace using the HollowNet architecture can lead to improvements on standard benchmark datasets, static MNIST and Omniglot. Furthermore, we can estimate the NLL of our models using $k = 5000$ for evaluating the quality of the generative model. Interestingly, although the NLLs were not improved significantly, CNFs can achieve much better ELBO values. We conjecture that the CNF approximate posterior may be slow to update, and has a strong effect of anchoring the generative model to this posterior.

### 4.3 Exact vs. Stochastic Continuous Normalizing Flows

We take a closer look at the effects of using an exact trace. We compare exact and stochastic trace CNFs with the same architecture and weight initialization. Figure 3 contains comparisons of models trained using maximum likelihood on the MINIBOONE dataset preprocessed by Papamakarios et al. (2017). The comparisons between exact and stochastic trace are carried out across two network settings with 1 or 2 hidden layers. We find that not only can exact trace CNFs achieve better training

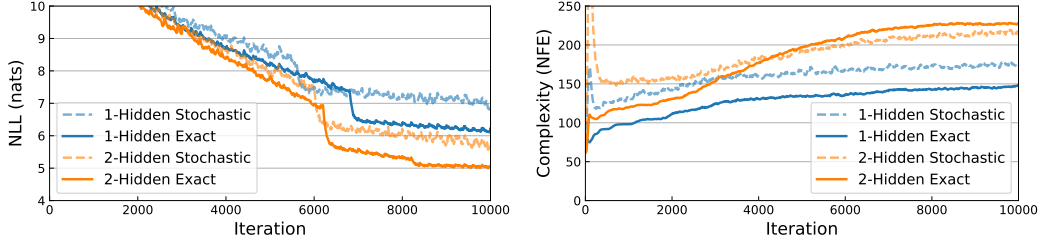

Figure 3: Comparison of exact trace versus stochastically estimated trace on learning continuous normalizing flows with identical initialization. Continuous normalizing flows with exact trace converge faster and can sometimes be easier to solve, shown across two architecture settings.

NLLs, they converge faster. Additionally, exact trace allows the ODE to be solved with comparable or fewer number of evaluations, when comparing models with similar performance.

## 5 Learning Stochastic Differential Equations by Fokker–Planck Matching

Generalizing ordinary differential equations to contain a stochastic term results in stochastic differential equations (SDE), a special type of stochastic process modeled using differential operators. SDEs are applicable in a wide range of applications, from modeling stock prices (Iacus, 2011) to physical dynamical systems with random perturbations (Øksendal, 2003). Learning SDE models is a difficult task as exact maximum likelihood is infeasible. Here we propose a new approach to learning SDE models based on matching the Fokker–Planck (FP) equation (Fokker, 1914; Planck, 1917; Risken, 1996). The main idea is to explicitly construct a density model $p(t, x)$, then train a SDE that matches this density model by ensuring that it satisfies the FP equation.

Let $x(t) \in \mathbb{R}^d$ follow a SDE described by a drift function $f(\mathbf{x}(t), t)$ and diagonal diffusion matrix $g(x(t), t)$ in the Itô sense.

$$dx(t) = f(x(t), t)dt + g(x(t), t)dW \tag{14}$$

where $dW$ is the differential of a standard Wiener process. The Fokker–Planck equation describes how the density of this SDE at a specified location changes through time $t$. We rewrite this equation in terms of $\mathcal{D}_{\text{dim}}^k$ operator.

$$\frac{\partial p(t, x)}{\partial t} = -\sum_{i=1}^{d} \frac{\partial}{\partial x_i} \left[ f_i(t, x)p(t, x) \right] + \frac{1}{2} \sum_{i=1}^{d} \frac{\partial^2}{\partial x_i^2} \left[ g_{ii}^2(t, x)p(t, x) \right]$$

$$= \sum_{i=1}^{d} \Bigg[ -(\mathcal{D}_{\text{dim}}f)p - (\nabla p) \odot f + (\mathcal{D}_{\text{dim}}^2 \text{diag}(g))p \tag{15}$$

$$+ 2(\mathcal{D}_{\text{dim}}\text{diag}(g)) \odot (\nabla p) + \nicefrac{1}{2}\,\text{diag}(g)^2 \odot (\mathcal{D}_{\text{dim}}\nabla p) \Bigg]_i$$

Written in terms of the $\mathcal{D}_{\text{dim}}$ operator makes clear where we can take advantage of efficient dimension-wise derivatives for evaluating the Fokker–Planck equation.

For simplicity, we choose a mixture of $m$ Gaussians as the density model, which can approximate any distribution if $m$ is large enough.

$$p(t, x) = \sum_{c=1}^{m} \pi_c(t)\mathcal{N}(x; \nu_c(t), \Sigma_c(t)) \tag{16}$$

Under this density model, the differential operators applied to $p$ can be computed exactly. We note that it is also possible to use complex black-box density models such as normalizing flows (Rezende and Mohamed, 2015). The gradient can be easily computed with a standard automatic differentiation, and the diagonal Hessian can be easily and cheaply estimated using the approach from Martens et al. (2012). HollowNet can be used to parameterize $f$ and $g$ so that the $\mathcal{D}_{\text{dim}}$ and $\mathcal{D}_{\text{dim}}^2$ operators operators in the right-hand-side of (15) can be efficiently evaluated, though for these initial experiments we used simple 2-D multilayer perceptrons.

**Training.** Let $\theta$ be the parameter of the SDE model and $\phi$ be the parameters of the density model. We seek to perform maximum-likelihood on the density model $p$ while simultaneously learning an SDE model that satisfies the Fokker–Planck equation (15) applied to this density. As such, we propose maximizing the objective

$$\mathbb{E}_{t,x_t \sim p_{\text{data}}}\left[\log p_\phi(t, x_t)\right] + \lambda \mathbb{E}_{t,x_t \sim p_{\text{data}}}\left[\left|\left|\frac{\partial p_\phi(t, x_t)}{\partial t} - \text{FP}(t, x_t|\theta, \phi)\right|\right|\right] \tag{17}$$

where $\text{FP}(t, x_t|\theta, \phi)$ refers to the right-hand-side of (15), and $\lambda$ is a non-negative weight that is annealed to zero by the end of training. Having a positive $\lambda$ value regularizes the density model to be closer to the SDE model, which can help guide the SDE parameters at the beginning of training.

This purely functional approach has multiple benefits:

1. No reliance on finite-difference approximations. All derivatives are evaluated exactly.
2. No need for sequential simulations. All observations $(t, x_t)$ can be trained in parallel.
3. Having access to a model of the marginal densities allows us to approximately sample trajectories from the SDE starting from any time.

**Limitations.** We note that this process of matching a density model cannot be used to uniquely identify *stationary* stochastic processes, as when marginal densities are the same across time, no information regarding the individual sample trajectories is present in the density model. Previously Ait-Sahalia (1996) tried a similar approach where a SDE is trained to match non-parameteric kernel density estimates of the data; however, due to the stationarity assumption inherent in kernel density estimation, Pritsker (1998) showed that kernel estimates were not sufficiently informative for learning SDE models. While the inability to distinguish stationary SDEs is also a limitation of our approach, the benefits of FP matching are appealing and should be able to learn the correct trajectory of the samples when the data is highly non-stationary.

## 5.1 Alternative Approaches

A wide range of parameter estimation approaches have been proposed for SDEs (Prakasa Rao, 1999; Sørensen, 2004; Kutoyants, 2013). Exact maximum likelihood is difficult except for very simple models (Jeisman, 2006). An expensive approach is to directly simulate the Fokker–Planck partial differential equation, but approximating the differential operators in (15) with finite difference is intractable in more than two or three dimensions. A related approach to ours is pseudo-maximum likelihood (Florens-Zmirou, 1989; Ozaki, 1992; Kessler, 1997), where the continuous-time stochastic process is discretized. The distribution of a trajectory of observations $\log p(x(t_1), \ldots, x(t_N))$ is decomposed into conditional distributions,

$$\sum_{i=1}^{N} \log p(x(t_i)|x(t_{i-1})) \approx \sum_{i=1}^{N} \log \mathcal{N}\big(x(t_i); \underbrace{x(t_{i-1}) + f(x(t_{i-1}))\Delta t_i}_{\text{mean}}, \underbrace{g^2(x(t_{i-1}))\Delta t_i}_{\text{var}}\big), \tag{18}$$

where we've used the Markov property of SDEs, and $\mathcal{N}$ denotes the density of a Normal distribution with the given mean and variance. The conditional distributions are generally unknown and the approximation made in (18) is based on Euler discretization (Florens-Zmirou, 1989; Yoshida, 1992). Unlike our approach, the pseudo-maximum likelihood approach relies on a discretization scheme that may not hold when the observations are sparse and is also not parallelizable across time.

## 5.2 Experiments on Fokker–Planck Matching

We verify the feasibility of Fokker–Planck matching and compare to the pseudo-maximum likelihood approach. We construct a synthetic experiment where a pendulum is initialized randomly at one of two modes. The pendulum's velocity changes with gravity and is randomly perturbed by a diffusion process. This results in two states, a position and velocity following the stochastic differential equation

$$d\begin{bmatrix} p \\ v \end{bmatrix} = \begin{bmatrix} v \\ -2\sin(p) \end{bmatrix} dt + \begin{bmatrix} 0 & 0 \\ 0 & 0.2 \end{bmatrix} dW. \tag{19}$$

This problem is multimodal and exhibits trends that are difficult to model. By default, we use 50 equally spaced observations for each sample trajectory. We use 3-hidden layer deep neural networks

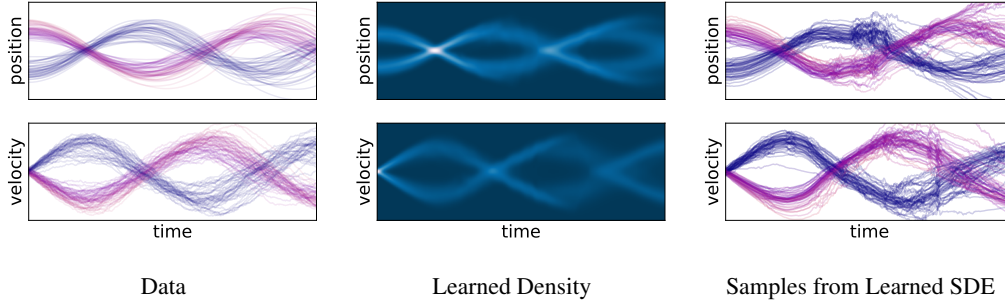

Data             Learned Density           Samples from Learned SDE

Figure 4: Fokker–Planck Matching correctly learns the overall dynamics of a SDE.

to parameterize the SDE and density models with the Swish nonlinearity (Ramachandran et al., 2017), and use $m = 5$ Gaussian mixtures.

The result after training for 30000 iterations is shown in Figure 4. The density model correctly recovers the multimodality of the marginal distributions, including at the initial time, and the SDE model correctly recovers the sinusoidal behavior of the data. The behavior is more erratic where the density model exhibits imperfections, but the overall dynamics were recovered successfully.

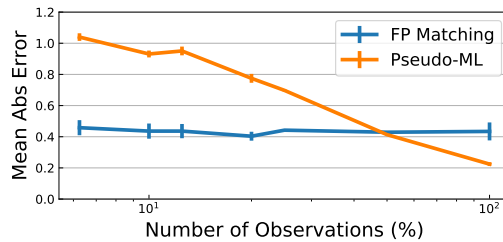

A caveat of pseudo-maximum likelihood is its reliance on discretization schemes that do not hold for observations spaced far apart. Instead of using all available observations, we randomly sample a small percentage. For quantitative comparison, we report the mean absolute error of the drift $f$ and diffusion $g$ values over 10000 sampled trajectories. Figure 5 shows that when the observations are sparse, pseudo-maximum likelihood has substantial error due to the finite difference assumption. Whereas the Fokker–Planck matching approach is not influenced at all by the sparsity of observations.

Figure 5: Fokker–Planck matching outperforms pseudo-maximum likelihood in the sparse data regime, and its performance is independent of the observations intervals. Error bars show standard deviation across 3 runs.

# 6   Conclusion

We propose a neural network construction along with a computation graph modification that allows us to obtain "dimension-wise" $k$-th derivatives with only $k$ evaluations of reverse-mode AD, whereas naïve automatic differentiation would require $k \cdot d$ evaluations. Dimension-wise derivatives are useful for modeling various differential equations as differential operators frequently appear in such formulations. We show that parameterizing differential equations using this approach allows more efficient solving when the dynamics are stiff, provides a way to scale up Continuous Normalizing Flows without resorting to stochastic likelihood evaluations, and gives rise to a functional approach to parameter estimation method for SDE models through matching the Fokker–Planck equation.

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
