[Reviews · NeurIPS 2019]

Reviewer 1



This paper proposes a neural network architecture that allows fast computation dimension-wise derivatives (regardless of dimensionality). Proposed modification in the computation graph leads to diagonal Jacobians (as well as higher order derivatives) that can be computed in a single pass. The model also generalizes a number of recently proposed flow models, which makes it a significant contribution to this line of research. The authors show that their idea can be combined with implicit ODE methods, which results in faster convergence rates in stiff dynamics. Continuous-time normalizing flows is also improved by computing the exact trace thanks to the proposed framework. Finally, DiffOpNet is proven useful for the inference of SDEs, which is known to be a difficult problem. The paper is very well-written and easy to follow, I really enjoyed reading the paper. The proposed idea seems to be pretty straightforward but also well-thought and developed throughout the paper. In addition to the idea, the paper is also novel in the sense that it brings together the recent advances in automatic differentiation and a neural network construction. I believe that we will be seeing more papers taking advantage of AD graphs in the near future. Hence, this paper, being one of the firsts, should definitely be accepted. Comments on the results: - Results in Figure1 is intuitive and really impressive but I do wonder how you identified the stiffness of the ODE systems. - I am particularly impressed by the improvement upon CNF in Table1. - Did you explore the SDE systems having d>2 dimensionality? - One restriction of the SDE inference is that the diffusion is diagonal. Would your framework still be used if it weren't? - Could you comment on the execution times? - Could you give a reference to your claim that setting k=5000 and computing (13) is the standard way of computing NLL? Replies to the above questions could also be included in the paper.

Reviewer 2



UPDATE: Many thanks to the authors for their response. I understand that training a neural ODE with an unconstrained network and exact trace evaluation is expensive in high dimensions. I remain positive about the paper, and I'm happy to keep my score of 8. I wish the authors best of luck and I look forward to see what comes next! Summary: The paper presents a new architecture for neural networks with D inputs and D outputs, called DiffOpNet. DiffOpNets have the property that differential operators such as the divergence or the Jacobian diagonal can be computed efficiently on them (whereas in general they'd need O(D) backprop passes). The paper presents three applications of DiffOpNets in solving, evaluating and learning differential equations; thanks to using DiffOpNets, the necessary differential operators can be efficiently computed. Originality: The idea of DiffOpNets is original and clever. DiffOpNets have some similarities to autoregressive transformations, which the paper clearly describes. The three applications of DiffOpNets (Jacobi-Newton iterations, exact log density of neural ODEs, learning SDEs) are original and exciting. Overall, the originality of the paper is high. Quality: The paper is of high technical quality, and includes sufficient mathematical detail to demonstrate the correctness of the proposed methods. I think a weakness of the paper is that it doesn't sufficiently evaluate extent to which the architectural constraints of DiffOpNets impact their performance. In particular, it would have been nice if section 4 contained an experiment that compared the performance (e.g. on density estimation) of a neural ODE that uses a DiffOpNet with a neural ODE that uses an unconstrained network of comparable size but computes the trace exactly. That way, we could directly measure the impact of the architectural constraints on performance, as measured by e.g. test log likelihood. Lines 61-63 compare the expressiveness of DiffOpNets to autoregressive and coupling transformations. As I understand it, the implied argument is that DiffOpNets impose fewer architectural constrains than autoregressive and coupling transforms, and since flows based on these transforms are expressive enough in practice, so should DiffOpNets. However I think this is a flawed argument: autoregressive and coupling transforms are composable, and even though each individual transform is weak, their composition can be expressive. On the other hand, it doesn't seem to me that DiffOpNets are composable; if we stack two DiffOpNets, the resulting model wouldn't have the same properties as a DiffOpNet. Section 5 evaluates Fokker-Planck matching is on a rather toy and very low-dimensional experiment (two dimensions). As I understand it, the main claim of that section is that Fokker-Planck matching with DiffOpNets scales better with dimensionality compared to other methods. Therefore, I think that I higher-dimensional experiment would strengthen this claim. Clarity: The paper is very well written, and I enjoyed reading it. Having the main method in section 2 and three stand-alone applications in sections 3, 4 & 5 works really well as a structure. Significance: I think DiffOpNets are a significant contribution, with various important applications in differential equations, as was clearly demonstrated. I think that the significance of the paper would have been higher if it included a clearer evaluation of the impact of the architectural constraints on the network performance. In any case I think that the paper deserves to be communicated to the rest of the NeurIPS community and I happily recommend acceptance.

Reviewer 3



The paper studied the problem of efficient derivative computation. The advantage is that the Jacobian is diagonal matrix for the given structure and the vector-Jacobian multiplication reduces to vector inner product. The major improvement of given method is computation efficiency. In the experiments, why there is no wall-clock time result to show such efficiency? Why an efficient gradient calculation can lead to better convergence point of the optimization objective? The paper is slightly out of the reviewer's area but the reviewer cannot get a clear understanding from the paper alone, probably due to some ambiguous notations: i) On page 2, it says setting h_i = x_\i can recover standard neural net, but doing so gives f_i = \tau_i(x) where standard neural net should be (f_1,…, f_d) = \tau (x) while here there are d networks? ii) What is g(x, h) on page 2? Is it \tau(x, h)? In general, the claimed improvements are not clearly reflected in the experiments and the improvement of writing is desired. ###### I have read the authors rebuttal and make the change of rating accordingly.

[Author Response · NeurIPS 2019]

**Reviewer #1:**  Thank you for your questions. We will incorporate our answers into the paper.

- We used a deep and thin network and a modified objective that emphasized the reconstruction aspect of the VAE. This led to very stiff models, but are of no practical use as they have to be trained for an extremely long time.

- We only explored 2d SDEs for this initial proof of concept as we found that the literature consists mostly of 2d synthetic SDEs when comparing different inference methods. We would appreciate it if the reviewer could suggest a suitable dataset if they have one in mind.

- With a full diffusion matrix of d*m, where m is the dimension of W, we would require m DiffOpNets, one for each row of the matrix. Our approach should still reduce computation by a factor of $d$, but we did not pursue this.

- The SDE training is very fast and can be trained in a few minutes due to its parallelism. The DiffOp-CNF is slower due to having to sequentially solve an ODE, and can take days for convergence on MNIST. Our method should be around the same time cost as FFJORD (which also reported to be slower than alternatives due to the sequential nature of solving an ODE).

- We will reference the original IWAE paper, "Importance Weighted Autoencoders" by Burda et al.

**Reviewer #2:**

**Comparing fairly to regular networks.**  This is difficult to carry out because (i) it's unclear how fair comparisons should be constructed and (ii) the exact dimension-wise derivative (as used for training e.g. CNFs) is only feasible in very low dimensions or otherwise would require a significant amount of time. For instance, FFJORD reported computing the exact trace to be infeasible and as such could not provide NLL estimates using the IWAE estimator. We did not come up with a satisfactory answer to this during the rebuttal, but we thank the reviewer for bringing this up.

**DiffOpNets are not composable.**  This is a very good point, and we will add this to the paper. For a function composition $f \circ g$, the Jacobian would be $J_f J_g$ whose diagonal cannot be computed using just the diagonals of each Jacobian. However, the networks that make up a DiffOpNet can have arbitrary depth and hidden state width.

We did not explore higher-dimensional SDE problems as we could not identify a suitable dataset, and we believe our experiment results would carry over to simple synthetic datasets. We would appreciate it if the reviewer could suggest a suitable dataset if they have one in mind. We hope to explore this combination of cheap derivatives and SDEs more in the future, as dimension-wise derivatives show up in many equations involving SDEs.

We also thank the reviewer for their minor corrections/suggestions.

**Reviewer #3:**  We thank the reviewer for taking the time to read our paper, and we apologize that the reviewer found our paper to be difficult to comprehend. We understand that we choose a non-standard format of having one core idea and three stand-alone applications which may as of now be only of a niche interest. However, we believe our contributions are useful to machine learning research, and we hoped to convey its wide applicability (ranging from numerical integration, to modeling, to optimization) using our format. We believe there are many areas and applications that can make use of dimension-wise derivatives but are yet unexplored due to computational infeasibility.

We plan on including more background information in each section using the extra page allowance should this paper be accepted. The reason we chose these applications is because the CNF experiments are on standard benchmark datasets, while the SDE experiments motivate the use of second-order dimension-wise derivatives.

Our main contribution lies in reducing the computational cost by a factor of $d$. Comparison is difficult, as naïve computation would be infeasible in but all but very low dimensions. Instead, we focused on motivating why computing the dimension-wise derivatives are useful in the first place as this has not been explored in previous works due to its computational cost. We compared against a *stochastic* trace estimator which has the same time complexity as our method (and is less general), but has higher variance gradients, which can impede optimization.

It was shown [1] that for ODE models, the number of function evaluations is linear with the wall-clock time. As such, we only reported number of function evaluations as this is more reproducible and comparable between implementations.

i) While we still have $d$ networks, our comment is only meant to illustrate that the expressiveness would be equal to a standard network, i.e. it can express the same set of functions. If the bottleneck is large enough, one could set $\tau_i(x)$ to simply be the $i$-th output of a standard network.

ii) You're correct that $g$ is $\tau$. This was missed during a change of notation. We will correct this.

[1] "Neural Ordinary Differential Equations" Chen et al. (2018)

[Meta-Review · NeurIPS 2019]

The paper proposes deep neural networks architectures that are designed to allow for an efficient evaluation of differential operators. The approach is simple, original and likely to trigger further innovations in the coming years. It will also provide a baseline for future methods based on other approaches to solving the same type of problems.